# Untargeted Metabolomic Characteristics of Skeletal Muscle Dysfunction in Rabbits Induced by a High Fat Diet

**DOI:** 10.3390/ani11061722

**Published:** 2021-06-09

**Authors:** Huimei Fan, Yanhong Li, Jie Wang, Jiahao Shao, Tao Tang, Mauricio A. Elzo, Li Wang, Tianfu Lai, Yuan Ma, Mingchuan Gan, Xianbo Jia, Songjia Lai

**Affiliations:** 1College of Animal Science and Technology, Sichuan Agricultural University, Chengdu 611130, China; fanhuimei1998@163.com (H.F.); lyh81236718@163.com (Y.L.); shaojh1997@163.com (J.S.); m18483220592@163.com (T.T.); wangsoli@163.com (L.W.); tf.lai@foxmail.com (T.L.); manima0916@163.com (Y.M.); ganmingchuan1998@163.com (M.G.); jaxb369@sicau.edu.cn (X.J.); laisj5794@163.com (S.L.); 2Department of Animal Sciences, University of Florida, Gainesville, FL 32611, USA; maelzo@ufl.edu

**Keywords:** rabbit, high-fat diet, skeletal muscle, metabolomics, biomarkers

## Abstract

**Simple Summary:**

In the present study, we performed an untargeted metabolomic analysis of skeletal muscle of rabbits and found that the skeletal muscle of rabbits fed a high-fat diet is rich in many metabolites, most of which are associated with type 2 diabetes and metabolic syndrome. In this paper, the mechanism of action of these metabolites in skeletal muscle and the metabolic pathways that interfere with the normal operation mechanism of the body are described and presented in the form of charts. Finally, we found that skeletal muscle-rich phospholipids, long-chain carnitine, histidine, carnosine, and tetrahydrocortisone may be potential markers for type 2 diabetes and metabolic syndrome, and may serve as potential therapeutic targets for related diseases in the future.

**Abstract:**

Type 2 diabetes and metabolic syndrome caused by a high fat diet (HFD) have become public health problems worldwide. These diseases are characterized by the oxidation of skeletal muscle mitochondria and disruption of insulin resistance, but the mechanisms are not well understood. Therefore, this study aims to reveal how high-fat diet causes skeletal muscle metabolic disorders. In total, 16 weaned rabbits were randomly divided into two groups, one group was fed a standard normal diet (SND) and the other group was fed a high fat diet (HFD) for 5 weeks. At the end of the five-week experiment, skeletal muscle tissue samples were taken from each rabbit. Untargeted metabolomic analysis was performed using ultra-performance liquid chromatography combined with mass spectrometry (UHPLC-MS/MS). The results showed that high fat diet significantly altered the expression levels of phospholipids, LCACs, histidine, carnosine, and tetrahydrocorticosterone in skeletal muscle. Principal component analysis (PCA) and least squares discriminant analysis (PLS-DA) showed that, compared with the SND group, skeletal muscle metabolism in HFD group was significantly up-regulated. Among 43 skeletal muscle metabolites in the HFD group, phospholipids, LCACs, histidine, carnosine, and tetrahydrocorticosteroids were identified as biomarkers of skeletal muscle metabolic diseases, and may become potential physiological targets of related diseases in the future. Untargeted metabonomics analysis showed that high-fat diet altered the metabolism of phospholipids, carnitine, amino acids and steroids in skeletal muscle of rabbits. Notably, phospholipids, LCACs, histidine, carnopeptide, and tetrahydrocorticosteroids block the oxidative capacity of mitochondria and disrupt the oxidative capacity of glucose and the fatty acid-glucose cycle in rabbit skeletal muscle.

## 1. Introduction

With the rapid development of the economy and society, great changes have taken place in the way of the human diet. It seems that the consumption of high-fat foods has become a normal part of life, which brings about many related diseases to the body. The most common is chronic metabolic disease, including obesity, hyperlipidemia, hypertension, and diabetes. The basic characteristics of these chronic diseases are alterations to the TCA cycle, blocked glucose-fatty acid cycle, elevated rates of lipolysis, and impaired glucose homeostasis, making the body unable to effectively absorb and store energy, resulting in metabolic disorders of the body [1,2]. However, diet is considered to be the most important factor leading to metabolic disorders [3,4].

A high-fat diet (HFD) causes obesity and metabolic diseases mainly due to the accumulation of metabolites in skeletal muscle. In addition to supporting movement and breathing, skeletal muscle is also the most important metabolic organ in the human body, which can regulate glucose homeostasis. At the end of a meal, 80% of the glucose is taken by skeletal muscles. Skeletal muscle not only plays an important role in glucose uptake, but it is also essential for ailments associated with metabolic disorders [5,6,7,8]. Over nutrition cause the deposition of metabolites in skeletal muscle, which leads to the interruption of the mitochondrial oxidation ability of skeletal muscle. The release of insulin in the muscle of the pancreas desensitizes and inhibits absorption, leading to insulin resistance [8,9,10].

Moreover, there are an increasingly larger number of reports indicating that a HFD can cause skeletal muscle metabolism disorders resulting in related diseases. Studies have shown that a high-fat diet can cause blood sugar disorders and accelerate the loss of calcium in skeletal muscles, known medically as sarcopenia [2,11]. The ability of skeletal muscle to absorb glucose decreased in mice fed a HFD, resulting in increased oxidation rate of fatty acids in skeletal muscle, leading to the accumulation of toxic metabolites and pro-inflammatory cytokines, and further aggravating insulin resistance [2]. A HFD can trigger the release of more pro-inflammatory cytokines in skeletal muscle, especially the harmful pro-inflammatory cytokine IL-6, whose overexpression leads to severe muscle atrophy and chronic, low-level skeletal muscle inflammation [2,12,13]. The accumulation of octanol and palmitoyl carnitine in skeletal muscle and the reduction of mitochondrial oxidative phosphorylation of glutamic acid/malic acid in mice fed with a HFD led to hyperleptinemia [2]. These results suggest that a HFD can indeed lead to skeletal muscle metabolic disorders and related diseases.

In this study, we investigate whether a HFD alters the skeletal muscle metabolism physiology in rabbits. We utilize a metabolomic approach to assess the type and number of metabolites in skeletal muscle as potential therapeutic targets to elucidate blocked metabolic pathways that may contribute to HFD associated metabolic diseases in rabbits such as obesity, type 2-diabetes, and hypertension. We hypothesized that feeding rabbits with a HFD for five weeks would change rabbit skeletal muscle metabolism physiology. Thus, the aim of this study was to determine whether rabbits in a HFD group altered their skeletal muscle metabolic physiology compared to rabbits in the normal-diet group to elucidate metabolites associated with skeletal muscle metabolic disorders.

## 2. Materials and Methods

### 2.1. Ethics Statement

This study was approved and conducted in accordance with the ethical standards of the Institutional Animal Care and Use Committee of the College of Animal Science and Technology, Sichuan Agricultural University, Sichuan, 611130, China.

### 2.2. Animals and Feeding Strategy

In total, 36 35-day-old female Tianfu black rabbits from Sichuan Agricultural University were selected and fed continuously for 5 weeks. At the start of the experiment, they were randomly divided into two groups. One group was fed a standard normal diet (SND), while the other group was fed a high-fat diet (HFD, with 10% lard added to the standard normal diet). Its SND was formulated in accordance with the nutritional requirements of the French INRA, and the nutritional composition is listed in the attached table (S1), which has been described in previous studies [1]. Feed 120 g twice a day. The water is provided free of charge and can be drunk freely. Each rabbit was kept separately in a clean cage (600 mm × 600 mm × 500 mm) and placed in an environmentally controlled room (21–23 °C, 60–75% humidity, 14-hourslight [60 lx]). As described in previous studies, daily food intake and weekly weight were recorded, as well as body length and chest length at the start and end of the trial. Obesity is defined based on these data [1].

### 2.3. Skeletal Muscle Tissue Collection and Preparation

At the end of the 35th day of the trial, eight rabbits in each group were screened out and sacrificed by intravenous injection. 16 samples of skeletal muscle tissue were immediately collected, freeze-dried, and crushed. Skeletal muscle tissue samples (100 mg) were individually grounded with liquid nitrogen and the homogenate was resuspended with prechilled 80% methanol and 0.1% formic acid by well vortex. The samples were incubated on ice for 5 min and then centrifuged at 15,000× *g*, 4 °C for 20 min. The supernatant was diluted with LC-MS grade water to a final concentration containing 53% methanol. The samples were subsequently transferred to a fresh Eppendorf tube and centrifuged at 15,000× *g*, 4 °C for 20 min. Finally, the supernatant was injected into the Liquid Chromatography with Tandem Mass Spectrometry (LC-MS/MS) analysis system [12]. Equal volume samples were taken from each test sample and mixed as quality control (QC) samples to ensure the robustness of the large-scale analysis.

### 2.4. Metabolomic Profiling

An untargeted metabolomics profiling was performed using ultraperformance liquid chromatography combined with mass spectrometry (UHPLC-MS/MS) with a Vanquish UHPLC system (ThermoFisher, Bremen, Germany) coupled with an Orbitrap Q ExactiveTM HF mass spectrometer (Thermo Fisher, Bremen, Germany) in Novogene Co., Ltd. (Beijing, China). Samples were injected into a Hypesil Goldcolumn (100 mm × 2.1 mm, 1.9 μm) using a 17-min linear gradient at a flow rate of 0.2 mL/min. The eluents for the positive polarity mode were eluent A (0.1% FA in Water) and eluent B (Methanol). The eluents for the negative polarity mode were eluent A (5 mM ammonium acetate, pH 9.0) and eluent B (Methanol). The solvent gradient was set as follows: 2% B, 1.5 min; 2–100% B, 12.0 min; 100% B, 14.0 min; 100–2% B, 14.1 min; 2% B, 17 min. The Q ExactiveTM HF mass spectrometer was operated in positive/negative polarity mode with a spray voltage of 3.2 kV, a capillary temperature of 320 °C, a sheath gas flow rate of 40 arb, and an aux gas flow rate of 10 arb.

### 2.5. Metabolite Recognition and Data Preprocessing

The raw data files generated by UHPLC-MS/MS were processed using Compound Discoverer 3.1 (CD3.1, ThermoFisher) to perform peak alignment, peak picking, and quantitation for each metabolite. The main parameters were set as follows: retention time tolerance, 0.2 min; actual mass tolerance, 5 ppm; signal intensity tolerance, 30%; signal/noise ratio, 3; and minimum intensity, 100,000. Subsequently, peak intensities were normalized to the total spectral intensity. The normalized data were used to predict the molecular formula based on additive ions, molecular ion peaks, and fragment ions. Peaks were matched with mzCloud (https://www.mzcloud.org/ (accessed on 4 December 2020)), mzVault, and MassList database to obtain accurate qualitative and relative quantitative results.

### 2.6. Statistical Analysis

Statistical analyses were performed using R statistical software (R version R-3.4.3), Python (Python 2.7.6 version), and CentOS (CentOS release 6.6). When data were not normally distributed, normal transformations were attempted using the area normalization method.

A principal component analysis (PCA) was used to assess the stability of the analytical process and to visualize global metabolome differences between SND and HFD rabbit groups. To maximize class discrimination necessary for discovery of potential metabolic biomarkers between SND and HFD rabbit groups, a supervised model of partial least squares discriminant analysis (PLS-DA) was applied. The variable importance in the projection (VIP) value was used to estimate the discriminatory power of each variable for the separation of the SND and HFD rabbit groups in the PLS-DA model, and variables with VIP > 1 were considered important. Furthermore, we performed a permutation test to ensure that the PLS-DA model was not over fitting.

For clustering heat maps, the data were normalized using z-scores of the intensity areas of differential metabolites and were plotted using a heatmap package in R. Pearson correlation coefficients between differential metabolites in the SND and HFD rabbit groups were analyzed in R with cor nd their significance tested with Test. A *p*-value < 0.05 was considered as statistically significant. Correlation plots were drawn with R program corrplot.

The functions of metabolites and metabolic pathways in the SND and HFD rabbit groups were studied using the KEGG database. Metabolites with VIP > 1, *p*-value < 0.05, and fold change ≥2 or FC ≤ 0.5 were considered to be differential metabolites. Volcano plots were used to filter metabolites of interest based on metabolite values of log2 (FoldChange) and -log10 (*p*-value).

## 3. Results

### 3.1. Effect of HFD on Skeletal Tissue Metabolomics

The PCA values (Figure 1a,b) and the PLS-DA results (Figure 1c,d) showed that the skeletal muscle of SND and HFD rabbit groups differed from each other. The values of the model quality parameters for skeletal muscle were R2 = 0.84 and Q2 = −0.81 for the positive model data (Figure 1c) and R2 = 0.70 and Q2 = −0.82 for the negative model data (Figure 1d). These results showed that the PLS-DA model had a good predictive ability and there was no overfitting.

Endogenous substances with VIP > 1 were selected to further analyze the effects of HFD on metabolic pathways in skeletal muscle. Table 1 shows that 43 endogenous metabolites had VIP > 1 in skeletal muscle, and that 26 of them had higher values and 17 had lower values in HFD than in SND rabbits. The levels of these 43 metabolites were visualized in a volcano figure (Figure 1e) and a heat map (Figure 1f). These figures show that HFD affects the metabolism of phospholipids, carnitine, amino acids, and steroids in the skeletal muscle of rabbits, and that the HFD greatest influence occurs on amino acid and lipid metabolism.

Endogenous substances with VIP > 1 were selected to further analyze the effects of HFD on metabolic pathways in skeletal muscle. Table 1 shows that 43 endogenous metabolites had VIP > 1 in skeletal muscle, and that 26 of them had higher values and 17 had lower values and (Table 2) n HFD than in SND rabbits. The levels of these 43 metabolites were visualized in a volcano figure (Figure 1e) and a heat map (Figure 1f) show that HFD affects the metabolism of phospholipids, carnitine, amino acids, and steroids in the skeletal muscle of rabbits, and that the HFD greatest influence occurs on amino acid and lipid metabolism.

### 3.2. Effects of HFD on the Metabolic Pathways of Endogenous Substances in Skeletal Muscle

The results showed that HFD affected 13 metabolic pathways related to endogenous substances in skeletal muscle (Figure 2). In addition, HFD affected endogenous substances related to metabolic pathways in rabbits, especially histidine and alanine metabolism. This indicates that HFD primarily affected amino acid metabolism in skeletal muscle of rabbits after 30 days of feeding with a HFD, leading to changes in endogenous substances. A metabolite synthesis diagram is shown in Figure 3.

## 4. Discussion

High-fat foods are harmful to rabbits because they contain large amounts of fats and oxides. They can destroy the ability of rabbit skeletal muscle to metabolize glucose and produce harmful substances that affect the normal growth and development. Results here showed that a HFD altered the metabolism of phospholipids, carnitine, amino acids, and steroids in the skeletal muscle of rabbits. Notably, there were significant changes in the expression levels of histidine, leucine, carnosine, and phospholipids in the HFD group. This was confirmed by the metabolomics analysis.

The lipids as a structural element of the cell membrane, lipids regulate metabolic homeostasis through various mechanisms, thus they are essential to maintaining homeostasis. However, lipids can also have deleterious effects on glucose metabolism and insulin sensitivity. When animals receive a HFD and obesity occurs, the body’s normal metabolism slows dramatically, and adaptive mechanisms often fail to give the body appropriate feedback. The resulting influx of lipids from adipose tissue exceeds its storage capacity, leading to the accumulation of harmful lipids in skeletal muscle, which is thought to be an important factor in insulin resistance [14]. Most phospholipids exist in skeletal muscle membranes and are important regulators of skeletal muscle mitochondrial respiratory function [15,16]. Relevant studies have shown that increased phospholipids in skeletal muscle may lead to changes in mitochondrial metabolism or increase in mitochondrial membrane remodeling, making phospholipids play a role in the inflammation of skeletal muscle. Further, excess lipids have harmful effects on the body through the action of the endoplasmic reticulum. In the case of a HFD or chronic nutritional stress, lipid synthesis in the endoplasmic reticulum is disordered, which interrupts the calcium signal [15]. Lysophosphatidic acid is a kind of lipid with strong biological activity produced by the ATX enzyme. High-fat-induced changes in LPA concentrations have been associated with the development of obesity-related damage to glucose homeostasis in addition to inflammatory disease [17,18]. Previous studies showed that excessive endogenous LPA production in HFD mice inhibited glucose tolerance, which may lead to the deterioration of glucose tolerance [19,20]. In this study, PC and PE (18:1) in the HFD rabbit group were significantly up-regulated, consistent with previous studies, and could be used as a potential biomarker for inflammation. However, different levels of lysophospholipids (LPC (15:1), LPC (16:1), LPC (17:1)) were significantly down-regulated compared with the SND rabbit group, contrary to previous studies. These results suggest that a HFD may not have adverse effects on LPA metabolism in rabbit skeletal muscle. In other words, not all high-fat diets have harmful effects, but they do promote skeletal muscle growth and development to some extent.

In this study, we found that acetyl carnitine was significantly up-regulated and propionylcarnitine and propionylcarnitine were significantly down-regulated in the HFD rabbit group compared with the SND rabbit group. Carnitine is a type B fatty acid produced during the esterification of fatty acids. It plays a vital role in the regulation of muscle oxidative metabolism. Ninety five percent of medium-chain acylcarnitines (MCACs) and long-chain acylcarnitines (LCACs), and intermediate accumulation of fatty acid oxidation in skeletal muscle may lead to insulin resistance. Long chain carnitine accumulates during over nutrition, fasting, heart disease and insulin resistance [21]. There is increasing evidence that LCACs refers to the metabolic syndrome caused by excess energy [22]. Feeding rats with a HFD reduces the availability of LCACs and inhibits metabolic flexibility [23]. As acetylcarnitine is mediated by carnitinyl transferase (CrAT), HFD reduces the LCACs activity in obese rats, aggravates the circulation of glucose-fatty acids, and leads to metabolic rigidity caused by diet [24]. Hexanoylcarnitine, propionylcarnitine and propionylcarnitine are the LCAC fatty acid derivatives of carnitine [25]. Hexanoylcarnitine was markedly increased in the HFD rabbit group than in the SND rabbit group in this study. This indicates that a LCACs increase in skeletal muscle can cause a disturbance in fatty acid-glucose circulation. This metabolism change of LCACs is helpful to improve our understanding of a potential physiological mechanism for skeletal muscle dysfunction in HFD rabbits. However, propionylcarnitine and DL-Carnitine decreased in the HFD group. High levels of propionylcarnitine and DL-Carnitine may serve as a marker of vitamin B12 deficiency, be associated with orofacial clefts, and have beneficial effects on cardiac metabolism, myocardial microvasculature function during ischemia, and coronary blood flow [26]. These two LCACs may not be harmful to the body. Although previous studies have shown that increased concentrations of these LCACs are beneficial to the body, the mechanism of the influence of low concentrations of LCACs on the body is still unclear and needs further study. Longer studies are needed to investigate if reductions of propionylcarnitine and DL-Carnitine levels could also decrease insulin sensitivity. The effect of LCACs on skeletal muscle metabolism in rabbits was described only by metabonomic methods in this paper, thus further studies on other aspects are needed.

Amino acids are a class of organic nitrogen sources closely related to carbohydrate metabolism, and their synthesis is closely related to degradation and the TCA cycle [2].f the skeletal muscle cell anabolic signaling pathway is abnormal, then the transmission of amino acids will be reduced, resulting in blocked synthesis and interruption of mitochondrial oxidation ability, which may cause diseases such as type 2 diabetes and insulin resistance. When dietary lipid exceeds demand, phosphorylation of nitric oxide synthase occurs in the skeletal muscle endothelium, resulting in reduced amino acid metabolism, leading to the accumulation of large amounts of amino acids in skeletal muscle [27]. Additionally, amino acids are closely linked to glucose metabolism in skeletal muscle. They provide raw material for gluconeogenesis through catabolism and make gluconeogenesis play an important role in glucose metabolism [28,29]. Branched chain amino acids (leucine, isoleucine, and valine), aromatic amino acids (phenylalanine, tyrosine and tryptophan), and aliphatic amino acid (lysine) are generated by carboxylic acid by transamination of intermediates dependent on glucagon and insulin secretion to regulate glucose metabolism [30,31]. Other studies reported that endogenous amino acids increase in skeletal muscle is related to diabetes, making amino acids a candidate biomarker for related diseases [30]. Due to the high levels of amino acids compete with glucose oxidation at the substrate level and interfere with the transcription of insulin signals, thus reducing the sensitivity of insulin signals and leading to impaired glucose oxidation. Altered levels of four amino acids were detected in rabbit skeletal muscle in this study using an untargeted metabolomics approach, namely leucine, glycine, lysine, and histidine. Leucine was significantly down-regulated whereas glycine, lysine, and histidine were significantly up-regulated in skeletal muscle from the HFD group compared to the SND group. Increased levels of branched chain amino acids (especially leucine) induced by high fat were found to promote insulin resistance by activating phosphorylation of mTORC1 and S6 kinases and insulin receptor substrates S1 and S2 [32]. Reduced levels of glycine and lysine in serum due to dietary changes are thought to increase insulin resistance, with a risk of inducing type 2-glycosuria [33]. Subsequently, studies have shown that glycine concentration reduction was a powerful indicator of future diabetes [34]. Skeletal muscle contains a large amount of carnosine, a histidine residue [35]. The ingested histidine is decomposed and converted into glutamate by the action of glutamate synthetase L-Glutamine [36]. Previous studies have shown that the body can maintain the balance of nitrogen by enhancing the catabolic pathways of hemoglobin and carnosine so that hemoglobin levels and carnosine levels in the blood decrease [37]. Histidine, carnosine, and glutamine play an important role in oxidative stress and immune response. Numerous studies have shown that the antioxidant activity of carnosine occurs through ROS/RAS clearance and high lipid oxidation. If the concentration of histidine is too high, a large amount of high nitrogen ammonia will be produced during the decomposition process, changing the anti-oxidation effect of carnosine, leading to accelerated lipid oxidation, and diseases related to insulin resistance [38]. In addition to increase in the production of ammonia, high concentrations of histidine will lead to changes in the concentration of several amino acids; glutamic acid, alanine and glutamine will increase, and branched amino acids (valine, leucine, and isoleucine) will decrease, leading to amino acidemia [39]. In this study, histidine and carnosine levels in the HFD group were significantly up-regulated compared to the SND group, leading to significantly up-regulated glutamine levels and significantly down-regulated leucine levels, consistent with previous reports. This suggests that a HFD has specific effects on skeletal muscle metabolism in rabbits, which can be used as potential markers for metabolic diseases. However, contrary to previous research, four consecutive weeks of HFD failed to cause metabolic disorders due to high levels of lysine and glycine in rabbit skeletal muscle, which further ensured the normal operation of TCA circulation, gluconeogenesis and oxidative decomposition of glucose, and maintained the normal growth and development of rabbits. Moreover, previous studies reported that elevated plasma leucine levels were associated with insulin-resistance related diseases, but excessive histidine in skeletal muscle could lead to decreased leucine levels and insulin-resistance related diseases. These contrasting results may be an indication that leucine content in different parts of the body will have a different action mechanism, which needs further study.

Glucocorticoids containing corticosterone, cortisol, and tetrahydrocorticosterone are produced by the adrenal glands [40]. As a substance of cholesterol metabolism, glucocorticoids are an important part of the biological stress response that helps regulate body balance and glucose homeostasis. Protein breakdown and lipid metabolism in skeletal muscle are essential for maintaining normal bone cell function [41]. With changes in diet, bone cells can adversely affect the skeletal system by altering the activity of glucocorticoid levels causing muscle wasting, osteoporosis, and insulin resistance [42]. A large number of studies have shown that secondary osteoporosis, muscle atrophy, insulin resistance, and type 2 diabetes caused by excessive glucocorticoid in skeletal muscle are mainly caused by high-fat diets [43]. For example, the increase of glucocorticoids in skeletal muscle caused by high-fat diets is due to the increase of free fatty acids in the blood circulation, which inhibits the synthesis of protein, resulting in a negative nitrogen balance in skeletal muscle, thus causing muscle atrophy [44,45]. Glucocorticoids function primarily through transcriptional regulation of metabolic genes such as glucose 6 phosphatase (G6Pase) and phosphoenolpyruvate carbox61 kinase (PEPCK), which act as rate limiting enzymes for gluconeogenesis. When mice were fed a high-fat diet, lipids were deposited in skeletal muscles, resulting in increased fat decomposition, gluconeogenesis disorder, and reduced stress response. This can lead to an increase in glucocorticoids followed by metabolic syndrome and type 2 diabetes [46]. Conversely, mice fed a high-fat diet for 8 weeks had similar levels of corticosterone as mice fed a low-fat diet [47]. In the present study, rabbits in the HFD group had significantly higher concentrations of tetrahydrocorticosterone than rabbits in the SND group, consistent with the above reports, indicating that tetrahydrocorticosterone could be used as a potential marker for related diseases in skeletal muscle. However, corticosterone concentration decreased significantly. Other studies showed that corticosterone, in addition to being a biological pacemaker, controls physiological processes in mammals including humans, and it can also regulate the daily circadian rhythm by adjusting the level of adrenocorticosterone controlled by the suprachiasmatic nucleus. In addition, overall corticosterone levels were higher at night than during the day because rabbits eat 20 to 25% more at night than during the day [47]. Therefore, the results of the current study differ from those of previous studies perhaps due to the decrease of corticosterone levels in skeletal muscle caused by the unique biological characteristics of rabbits and the daytime samples taken in this trial. The level of glucocorticoids in skeletal muscle contributes to our understanding of steroid hormones, but the biological mechanism needs further research.

## 5. Conclusions

A high-fat diet caused a disturbance of the phospholipid metabolism, acylcarnitine metabolism, amino acid metabolism, and steroid metabolism in skeletal muscle of rabbits. The expression levels of phospholipids were significantly up-regulated, disrupting the oxidative ability of mitochondria, and impairing glucose homeostasis, potentially leading to inflammation-related diseases. Acetylcarnitine expression was significantly up-regulated, leading to insulin resistance and impaired glucose homeostasis, which may lead to chronic metabolic diseases such as type 2 diabetes, obesity, and hypertension. The expression levels of glycine and carnosine were significantly up-regulated, leading to excessive oxidative stress, lipid oxidation, insulin resistance, and disruption of the TCA cycle, which may be associated with hyperlipidemia, aminophenemia, and type 2-diabetes. The expression level of tetrahydrocortisol increased significantly, inhibiting protein synthesis and gluconeogenesis, which may lead to metabolic syndrome and type 2 diabetes. In summary, phospholipids, LCACs, histidine, carnosine, and tetrahydrocorticosterone could be used as potential biomarkers associated with chronic metabolic diseases, contributing to a better understanding of the underlying biological mechanisms, and providing a basis for the treatment and diagnosis of related diseases in the future.

## Figures and Tables

**Figure 1 animals-11-01722-f001:**
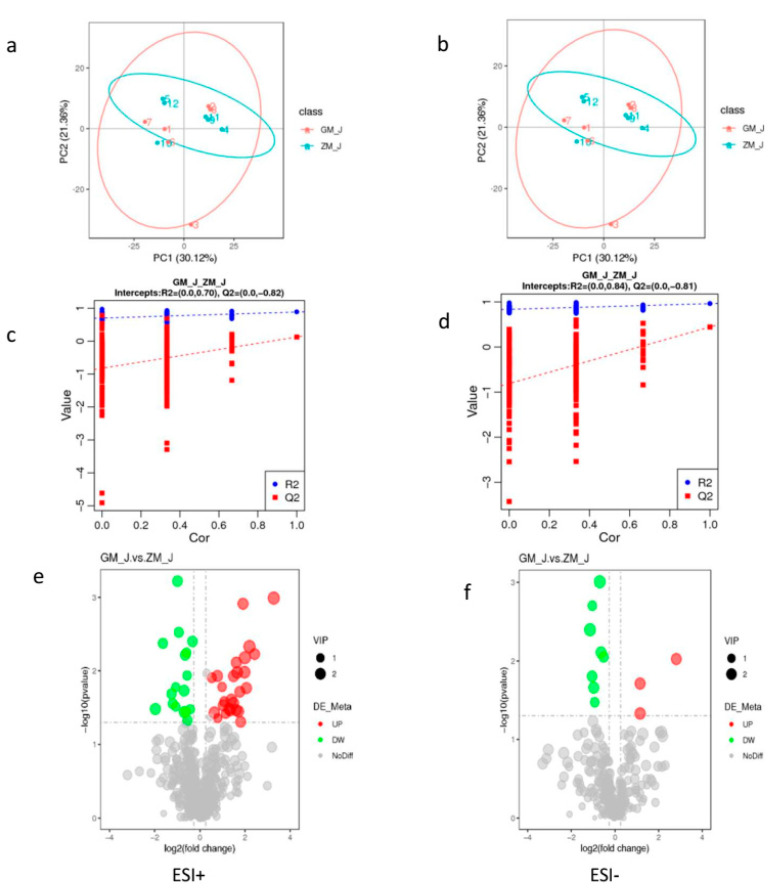
Skeletal muscle metabolite score and permutation test of rabbits in normal and high-fat diet groups. (**a**,**b**), PCA score graph between normal and high-fat diet groups; (**c**,**d**), permutation test from PLS-DA models; (**e**,**f**), volcano of metabolites with significant differences between normal and the high-fat diet groups; (**g**) is the clustering heat map of differential metabolites between normal and the high-fat diet groups. **Note:** ESI+: cationic mode; ESI-: Anion mode; GM: High fat group; ZM: Normal group.

**Figure 2 animals-11-01722-f002:**
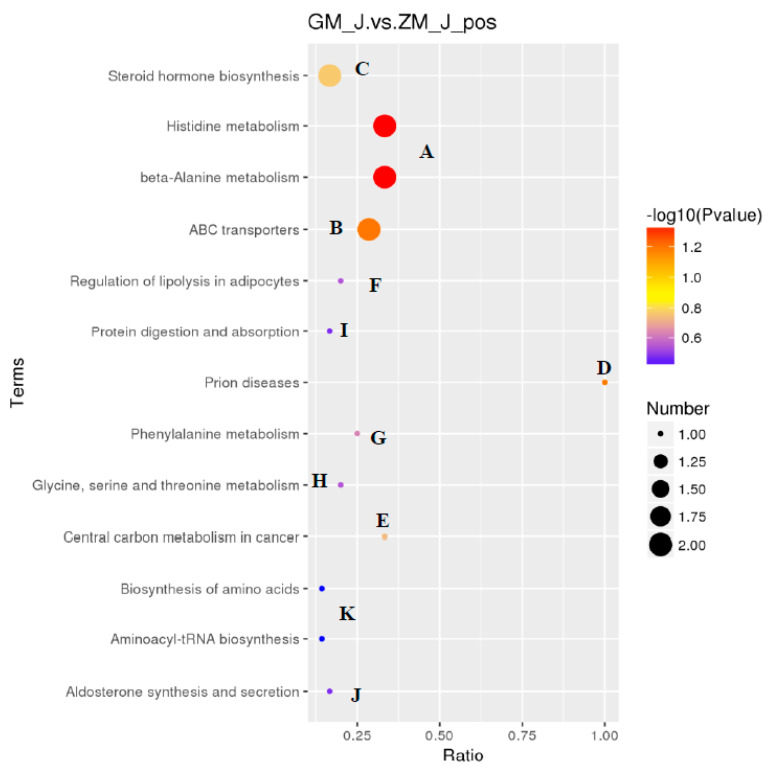
Disturbed pathways in response to a high-fat diet. A. Histidine metabolism and beta-Alanine metabolism; B. ABC transporters; C. Steroid hormone biosynthesis; D. Prion diseases; E. Central carbon metabolism in cancer; F. Regulation of lipolysis in adipocytes; G. Phenylalanine metabolism; H. Glycine, serine and threonine metabolism; I. Protein digestion and absorption; J. Aldosterone synthesis and secretion; K. Aminoacyl-tRNA biosynthesis and biosynthesis of amino acids.

**Figure 3 animals-11-01722-f003:**
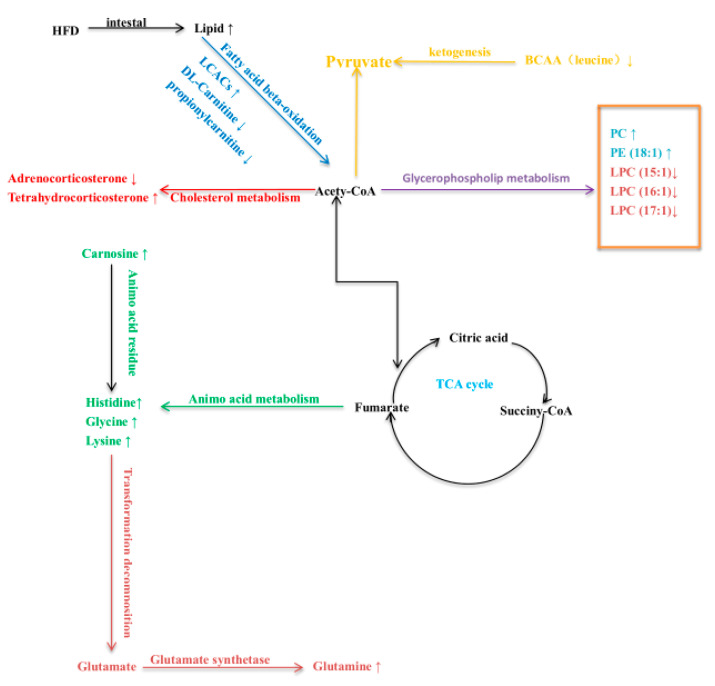
Interference and change of potential metabolic pathways in skeletal muscle of rabbits fed a high-fat diet. Different colors represent different metabolic pathways.

**Table 1 animals-11-01722-t001:** Composition and nutrient content of the standard normal diet (SND) and the high-fat diet (HFD) [13].

Ingredient	SND							HFD						
	Proportion (%)	DE (MJ/kg)	CP (%)	EE (%)	CF (%)	Ca (%)	P (%)	Proportion (%)	DE (MJ/kg)	CP (%)	EE (%)	CF (%)	Ca (%)	P (%)
Straw powder	26	0.855	1.248	0.364	7.748	0.073	0.021	23.5	0.773	1.128	0.329	7.000	0.066	0.019
Maize	18	2.889	1.602	0.648	0.576	0.005	0.070	16	2.568	1.424	0.576	0.512	0.004	0.062
Barley	20	2.808	2.040	0.34	0.860	0.020	0.093	18	2.527	1.836	0.306	0.774	0.018	0.084
Bran	15	1.631	2.310	2.475	0.765	0.050	0.072	13.5	1.468	2.079	2.228	0.689	0.045	0.065
Bean cake	16	2.166	6.768	0.304	0.576	0.045	0.091	14.5	1.963	6.134	0.276	0.522	0.041	0.082
Fish meal	3.5	0.552	2.047	0.196		0.137	0.104	3.15	0.497	1.842	0.176		0.123	0.094
Lard								10	3.683		9.8			
Stone powder	1.0					0.350		0.9					0.315	
Salt	0.5							0.45						
Total	100	10.91	16.015	4.327	10.525	0.68	0.431	100	13.479	14.443	13.691	9.497	0.621	0.406

**Table 2 animals-11-01722-t002:** Potential skeletal muscle biomarkers based on UPLC-Q-TOF/MS in rabbits fed a high-fat diet.

Name	Formula	RT [min]	m/z	*p*-Value	VIP	Trend
PC (14:1e/3:0)	C25 H50 N O7 P	14.594	508.33981	0.00060252	1.96834418	↓
ACar 10:2	C17 H30 N O4	10.771	312.21707	0.001022871	2.78164921	↑
Hexanoylcarnitine	C13 H25 N O4	8.781	260.18542	0.001221548	2.134885168	↑
Phenylacetylglycine	C10 H11 N O3	8.22	194.08138	0.002987006	1.562157041	↓
LPC 15:1	C23 H46 N O7 P	14.323	480.30856	0.004218472	1.679235509	↓
Palmitoylcarnitine	C23 H45 N O4	13.56	400.34161	0.004662458	2.728685353	↑
ACar 22:6	C29 H46 N O4	13.404	472.34155	0.005910097	2.009955994	↑
5-chloro-2,8-dimethyl-4-[(3-nitro-2-pyridyl)oxy]quinoline	C16 H12 Cl N3 O3	1.602	330.05746	0.006079831	1.752888258	↓
ACar 18:1	C25 H48 N O4	13.664	426.35709	0.006620219	2.680982815	↑
ACar 7:0	C14 H28 N O4	9.819	274.20126	0.007692279	2.101077514	↑
ACar 17:2	C24 H44 N O4	13.239	410.32602	0.010393589	2.48157315	↑
ACar 18:2	C25 H46 N O4	13.426	424.34134	0.010520878	2.366450915	↑
Ala-Gln	C8 H15 N3 O4	1.305	218.11295	0.011721656	2.174643548	↑
ACar 16:1	C23 H44 N O4	13.29	398.32599	0.011757863	2.207576851	↑
N-Acetyl-L-carnosine	C11 H16 N4 O4	1.381	269.12411	0.012378517	1.528953387	↑
LPC 22:6	C30 H50 N O7 P	14.45	568.33807	0.016506546	1.321877932	↑
PC (14:1e/2:0)	C24 H48 N O7 P	14.352	494.32413	0.016550657	1.102232178	↓
ACar 20:5	C27 H44 N O4	13.206	446.32581	0.017118755	2.11188684	↑
Muscone	C16 H30 O	14.308	239.23694	0.018553432	1.998565512	↓
ACar 20:4	C27 H46 N O4	13.413	448.3414	0.019178682	2.014369263	↑
Propionylcarnitine	C10 H19 N O4	2.882	218.13879	0.020501301	1.659387379	↓
ACar 17:1	C24 H46 N O4	13.479	412.34155	0.024652794	2.259139096	↑
ACar 20:3	C27 H48 N O4	13.578	450.35712	0.027362927	2.267210155	↑
LPC 15:0	C23 H48 N O7 P	14.46	482.32388	0.027825481	1.548906089	↓
ACar 15:1	C22 H42 N O4	13.095	384.3103	0.029353037	2.072617982	↑
LPE 17:0	C22 H46 N O7 P	15.085	468.30875	0.030453786	1.236219	↓
Corticosterone	C21 H30 O4	11.761	347.22134	0.032938755	2.163429757	↓
ACar 15:0	C22 H44 N O4	13.365	386.3259	0.032970783	2.228659726	↑
1-Palmitoyl-Sn-Glycero-3-Phosphocholine	C24 H50 N O7 P	13.491	496.33948	0.033163212	1.248603083	↓
ACar 18:3	C25 H44 N O4	13.203	422.32593	0.034156704	2.196673054	↑
ACar 12:1	C19 H36 N O4	12.16	342.26349	0.034297235	2.186103895	↑
Tetrahydrocorticosterone	C21 H34 O4	12.513	368.27911	0.035427687	2.2470103	↑
Prolylleucine	C11 H20 N2 O3	2.098	229.15422	0.035890864	1.53343442	↓
DL-Carnitine	C7 H15 N O3	1.31	162.11217	0.036302614	2.164262278	↓
L-Histidine	C6 H9 N3 O2	1.619	156.07664	0.036883235	2.296408916	↑
Carnosine	C9 H14 N4 O3	1.602	227.11374	0.038157621	1.374481323	↑
Lysops 22:5	C28 H46 N O9 P	14.594	572.29755	0.043926394	1.145501212	↑
Betaine	C5 H11 N O2	1.32	118.0863	0.046878628	1.626178977	↓
ACar 18:0	C25 H50 N O4	13.889	428.37296	0.049277304	1.980841845	↑

## Data Availability

All the figures and tables used to support the findings of this study are included within the article.

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
