# Peer review of "Untargeted Metabolomic Characteristics of Skeletal Muscle Dysfunction in Rabbits Induced by a High Fat Diet"

_animals, 2021, doi:10.3390/ani11061722_

Round 1

Reviewer 1 Report

IMPORTANT: In order to make a correct discussion and analysis of the work, I need the diets to be specified. It refers to the diets of another job, but they have to be present in this one, since it is a fundamental part. Without this information I cannot do a thorough review properly. The information of animals and feeding strategy should be better specified. in nutrition work, diets are the most important thing. Chemical composition and Ingredients.

On the other hand, I have found some other form improvements:

  • Figures must be self-explanatory, and any abbreviations must be defined in each table / figure.

  • For example, in figure 2 it has different colors, why are they?

  • Figure 1 should be divided into both

  • A guide to abbreviations used would improve understanding of the text.

  • Similar body weight? Please explain the main and SD.

  • Fed continuously? Under what kind of restriction?

  • N = 16? Do you think that is a representative number?

  • L98 is not justified.

Author Response

1.Dear teacher, I am very glad to get your advice.The animal information and feeding strategy you proposed, as well as the chemical composition of the feed, are all related to the papers published by other students in the research group.Moreover, his paper laid a certain foundation for my paper, so it was not mentioned in detail in my manuscript.And I am sending you the article as an attachment.

2.I will modify the picture again.(Note: ESI+ : cationic mode; ESI- :Anion mode; GM: High fat group;  ZM: Normal group.)

3.The relevant contents are in the attached results, and I only quote them in this manuscript..If there is any problem, I will correct it.

4.They were fed continuously for 35 days.Each rabbit was kept separately in a clean cage (600 × 600 × 500 mm) and placed in an environmentally controlled room (21–23°C, 60%–75% humidity, 14-hourslight [60 lx]).

5.I'm sorry about that.N should be the number of groups, and the total is 16.

Reviewer 2 Report

the rabbit reacts very quickly to stressful stimuli. In assessing the welfare of rabbits, physiological indicators such as cortisol and glucose levels are used. The size of the reaction is measured, among others, by the level of secreted glucose. The measurement of the concentration of hydrogen ions (pH) or the color of the meat is used as a parameter for assessing meat quality. The decrease in the pH of the meat proves that the glycolysis process is progressing correctly. Due to the high sensitivity of the rabbits, the respiratory action is accelerated and the parameters are significantly different from the norm. It is an animal species that is very sensitive and difficult to research. So why was the rabbit chosen and not, for example, pigs, which in many respects is very similar to humans? The very small number of animals used in the research, the unknown number of replications, do not allow such far-reaching conclusions. Other comments and questions are marked in the text of the publication. 

Author Response

Dear teacher, thank you very much for your questions and suggestions.As you said, the rabbit is a sensitive and very timid animal.However, it is often used in large numbers in clinical trials.And it is precisely because of its sensitive biological characteristics, as well as the differences and specificities between individuals, that it can be better used as a model to solve various human diseases.

Reviewer 3 Report

In my opinion, manuscript entitled ,,Untargeted metabolomic characteristics of skeletal muscle dysfunction in rabbits induced by a high fat diet” is well written and deserve to be published in ANIMALS.

My only one remark concerns line 93: BW index? I think that rabbits were weighted and those with similar BW were choosen. The word ,,index” means that to give an index, calculations from several other numbers are needed… So authors should use the term of BW, without ,,index”.

Congratulations to the authors of such well-planned and skilled work. Perhaps in the future it would be good to consider a similar experiment on rabbits using large amounts of vegetable fat, e.g. soybean or lenseed?

Author Response

Dear teacher, thank you for your recognition and advice.I am also very happy and honored to use this manuscript as a reference for my peersAs for the BW index you proposed, I did not mention it in the manuscript.It is because some students in the same laboratory did special research on this part of content in a paper before, and got certain results, so it is not mentioned in this manuscript.I'm very sorry for my negligence, so that the teacher had doubts.Here, I will send you the attachment, which is mentioned in detail in the result section.Moreover, I will revise and quote in detail in this manuscript.

Round 2

Reviewer 1 Report

I regret to insist that in my opinion, a nutrition work should refer to diet in it, and not have to make the reader take a paper and search the attached documents. (which do not appear in the document that they have sent me).

Author Response

This manuscript is a resubmission of an earlier submission. The following is a list of the peer review reports and author responses from that submission.